# An NLP-based novel approach for assessing national influence in clause dissemination across bilateral investment treaties

**Shahadat Uddin**[1]*, **Haohui Lu**[1], **Wolfgang Alschner**[2], **Dori Patay**[3], **Nicholas Frank**[4], **Fabio S. Gomes**[5], **Anne Marie Thow**[3]

**1** School of Project Management, Faculty of Engineering, The University of Sydney, Forest Lodge, Australia, **2** Common Law Section, Faculty of Law, University of Ottawa, Ottawa, Canada, **3** Sydney School of Public Health, Menzies Centre for Health Policy and Economics, Faculty of Medicine and Health, The University of Sydney, Forest Lodge, Australia, **4** School of Regulation and Global Governance, Australian National University, Canberra, Australia, **5** Department of Noncommunicable Diseases and Mental Health, Pan American Health Organization/World Health Organization, Washington DC, United States of America

* shahadat.uddin@sydney.edu.au

**Data Availability Statement:** The data used in this study are publicly available (https://edit.wti.org/document/investment-treaty/search).

## Abstract

International investment agreements (IIAs) promote foreign investment. However, they can undermine crucial health programs, creating a dilemma for governments between corporate and public health interests. For this reason, including clauses that safeguard health has become an essential practice in IIAs. According to the current literature, some countries have played a pivotal role in leading this inclusion, while others follow the former ones. However, the existing literature needs a unique approach that can quantify the influence strength of a country in disseminating clauses that explicitly mention health provisions to others. Following an NLP (Natural Language Processing)-based text similarity analysis of Bilateral Investment Treaties (BITs), this study proposes a metric, 'Influence' (*INF*), which provides insights into the role of different countries or regions in the propagation of IIA texts among BITs. We demonstrate a comprehensive application of this metric using a large agreement dataset. Our findings from this application corroborate the evidence in the current literature, supporting the validity of the proposed metric. According to the *INF*, Germany, Canada, and Brazil emerged as the most influential players in defensive, neutral, and offensive health mentions, respectively. These countries wield substantial bargaining power in international investment law and policy, and their innovative approaches to BITs set a path for others to follow. These countries provide crucial insights into the direction and sources of influence of international investment regulations to safeguard health. The proposed metric holds substantial usage for policymakers and investors. This can help them identify vital global countries in IIA text dissemination and create new policy guidelines to safeguard health while balancing economic development and public health protection. A software tool based on the proposed *INF* measure can be found at https://inftool.com/.

**Funding:** This study was funded by the National Health and Medical Research Council (NHMRC, Government of Australia), Grant ID 2012233. The funders had no role in study design, data collection and analysis, publication decisions, or manuscript preparation.

**Competing interests:** The authors have declared that no competing interests exist.

## 1. Introduction

Governments sign Bilateral Investment Treaties (BITs) to incentivise foreign investment as part of broader economic policy objectives. These agreements are negotiated between two governments to encourage, promote, and protect investments made by enterprises based in one country within the territory of the other [1]. However, these agreements have sometimes constrained policy space for vital health initiatives, such as tobacco control and pharmaceutical accessibility [2]. If not adequately protected, encouraging investment through BITs can constrain health policy [2]. In response, there has been increasing attention by the public health community regarding the importance of including clauses in BITs and other agreements that safeguard health and health policy-making [3].

Clauses safeguarding health have taken different forms, reflecting diverse priorities and their public health purpose. Thow et al. [3] classified the health safeguard into defensive, neutral and offensive categories. *Defensive* clauses focus on maintaining the integrity of existing health-related policies. Rather than advancing or expanding these policies, a defensive clause is primarily concerned with preserving the existing health-related policy space. This could involve offering clarifications, making exceptions, or implementing measures to ensure that the current levels of health protection are not compromised or lowered for other goals. It underlines a solid commitment to prioritising health regulations and preventing potential dilution or weakening of health policy frameworks. *Offensive* clauses refer to a proactive and assertive approach aiming to advance health policies. These clauses may include articulating expectations regarding investors' health-related responsibilities and using BITs as tools to advance and promote a positive public health agenda. There are also health clauses that neither serve as a defensive mechanism nor promote a progressive health agenda. Instead, these BITs uphold existing health policy norms and are therefore classified as '*neutral*' with respect to their health policy purpose. Health safeguards in BITs have traditionally been employed as a '*shield*' (defensive) to protect public health regulation. However, an emerging trend, particularly among developing countries, is using offensive clauses more proactively to direct foreign investment towards health projects and manage investors' health-related responsibilities.

Over time, there has been notable consistency in the wording of health safeguards [3]. That, in turn, suggests that states look to and learn from each other as they contemplate how to address health concerns in their agreements. The texts and innovative elements in early BITs frequently inspire clauses that other countries emulate in later treaties [4]. Understanding which countries play a significant role in disseminating BIT text can shed light on global economic governance and trade relations dynamics. Lie [5] also employs computational analysis to examine the evolution of international investment treaty language, revealing a dominant Western European influence, an early mover advantage in language propagation, and the nuanced role of negotiation power versus persuasive legal wording. These insights help track the propagation of diverse policy concepts, norms, and standards across various locations. Countries with strong bargaining power can establish standards that influence BIT text in other countries. Identifying these countries can also predict trends in international investment law and policy, aiding in comparing investment protection norms globally [6].

In examining the balance between International Investment Agreements (IIAs) and domestic health protections, this research echoes key insights from Voon and Mitchell [7]. Their detailed study of the Hong Kong–Australia BIT sheds light on the complex interplay between health-related clauses in BITs and broader legal and policy frameworks, an aspect crucial to our research. Chaisse [8] further contributes to this narrative by providing a comprehensive theoretical foundation and comparative analysis. His exploration into the propagation and implications of general exception clauses in BITs enhances our understanding of their

integration and impact. Additionally, the legal and policy perspectives surrounding tobacco regulation [9] enrich the discourse on the nuances of international investment law in public health. In addition, Vytiaganets [10] delves into tobacco regulation, offering critical insights into how BITs interact with and impact tobacco control measures.

Similarly, Melillo [11] discussed evidentiary issues in investment law, providing a deeper understanding of how evidence is handled in health and investment disputes. These studies collectively offer a comprehensive perspective on the legal complexities of international investment and public health policy, particularly in tobacco control. Chaisse et al. [8] also discussed Asian BIT practices and their influence on the Regional Comprehensive Economic Partnership (RCEP), with a specific focus on health-related exception clauses. Their analysis of the impact of Japanese agreements, like the Trans-Pacific and Japan-Uruguay BIT, on the RCEP investment chapter highlights the prevalence of language duplication and modification in treaty drafting. This is especially relevant to understanding the regional trends in Asia-Pacific, where health-related exceptions in BITs are increasingly critical in the backdrop of numerous tobacco-related disputes. Overall, through engagement with these pivotal studies, our research not only aligns with but also expands upon the current academic conversation, highlighting the dynamic nature and significant influence of health clauses in BITs.

This article introduces a novel metric, 'Influence' (*INF*), designed to capture a country's role in disseminating BIT clauses over time. The *INF* value represents influence: a higher value indicates a more significant impact held by a particular country/state in transferring BIT clauses, while a lower value suggests less influence. We use Natural Language Processing (NLP) techniques to analyse text similarity between BITs, generate *INF* for various countries, and compare these values. This allows us to identify key countries that have consistently played significant roles.

## 2. Proposed approach for assessing national influence

Any given BIT contains a range of different clauses. Countries often copy existing clauses when negotiating new BITs [12], leading to varying extents of similarity, from no (0%) to complete emulation (100%), between the clauses of any pair of BITs. Some clauses are reused more often than others. Some countries employ commonly used clauses for new BITs, while others consider rare ones. At the same time, governments tend to select clauses from their previous BITs [3]. These factors lead countries to play different roles in clause dissemination. Our basic assumption is that a country plays a significant role in disseminating clauses if the clauses of its BITs show high similarity with other BITs and vice versa. This study's proposed *INF* measure can capture countries' roles in disseminating clauses over time. The higher the value for this measure is, the more substantial the role played by the respective government and vice versa.

Each BIT between two countries or regions provides the following information—country or region names, the date when the agreement was signed, the agreement's main clause (document), and the category and subcategory based on the clause. This information is used in quantifying the INF value. The following are the steps for capturing countries' INF values considering all BITs for an underlying category/subcategory in a given period. Among these INF values, the country with the highest value is assumed to play the most significant role in disseminating clauses during that period for the underlying category/subcategory.

### Step 1: Text similarity

For the clauses of given BIT dyads (corpus), this step calculates the similarity between each pair of BITs. A document is an individual clause associated with a single BIT. This study uses the same ID for the document and BIT of an agreement. For comparing word similarity, this study

employs Term Frequency-Inverse Document Frequency (TF-IDF) [13] and cosine similarity [14]. The first method (TF-IDF) assigns a feature vector to each clause, and cosine similarity calculates the similarity between these feature vectors. Term Frequency-Inverse Document Frequency (TF-IDF) is a method used to quantify the significance of a word within a document that is part of a more extensive collection or corpus [15]. The formula for TD-IDF is -

$$w_{i,j} = TF_{i,j} \times log\left(\frac{N}{DF_i}\right) \tag{1}$$

Where $w_{i,j}$ is the weight of term $i$ in document $j$, $TF_{i,j}$ is the number of occurrences of term $i$ in document $j$, $N$ is the total number of documents and $DF_i$ is the number of documents containing the term $i$. The first part of the right-hand side of the above equation presents the TF, and the second represents the *IDF*.

In other words, the Term Frequency (TF) refers to the frequency with which a word appears in a specific document, also called the "bag of words" representation. The Inverse Document Frequency (IDF) represents the word frequency occurring across a collection of documents, commonly referred to as a corpus. The advantage of TF-IDF is its simplicity and efficiency.

Techniques like cosine similarity are frequently used, especially with TF-IDF vectorisation [16]. This method computes the cosine of the angle between two vectors while expressing the texts in a multidimensional space, allowing it to effectively assess similarity in big text datasets. If the corpus consists of $N$ BITs, then there will be $^NC_2 = \frac{N(N-1)}{2}$ text-similarity values. While the TF-IDF method offers simplicity and efficiency as its key advantages, it is imperative to consider the unique nature of legal documents. In legal texts, word order can be crucial for capturing clauses' precise meaning and nuance. As such, while TF-IDF excels in identifying term significance, it may not fully account for the structured intricacy of legal language. Therefore, it is worth acknowledging this limitation when applying the method to legal documents.

**TF-IDF example.**   In an effort to calculate the relevance of words within a series of texts, a simplified approach using the TF-IDF method was explored. Two sentences were considered: *D1*: *Investment agreements should protect health-related measures. D2*: *Investment agreements should protect climate-related measures.* Each of these documents consists of six words. The corpus (*Cor*) consists of *D1* and *D2*. To calculate the TF-IDF score of the word 'protect' in document D1,

$$TF(\,'protect',D1) = \frac{Number\ of\ times\ term\ 'protect'\ appears\ in\ D1}{Total\ number\ of\ terms\ in\ D1}$$

First, we see how often our target word, in this case, '*protect*', appears in the first sentence D1. It appears once, resulting in TF in a fraction of 1/6 or 0.167.

$$IDF\,(\,'protect',\ Cor\,) = log\left(\frac{Total\ number\ of\ documents}{Number\ of\ documents\ with\ term\ 'protect'\ in\ it}\right)$$

Next, we determine how many of our two sentences mention the word 'protect'. All two sentences use it, giving a fraction of IDF as 2/2 or 1. A mathematical operation called '*logarithm*' is applied to this fraction and becomes 0. The importance score for the word '*protect'* in the first sentence is found by multiplying the two numbers, resulting in 0.

Similarly, '*agreement*' appears once in D1, resulting in a TF value of 0.167, and '*agreement*' is also in both documents. Its IDF is $log\left(\frac{2}{2}\right) = 0$. Thus, when focusing on the terms '*protect*' and '*agreement*' for D1 and D2:

Vector for D1: [0, 0]

Vector for D2: [0, 0]

$$Cosine\ similarity = \frac{D1 \cdot D2}{\parallel D1 \parallel \parallel D2 \parallel}$$

Where $D1 \cdot D2$ is the dot product of the two vectors. $\parallel D1 \parallel$ is norm of D1 and $\parallel D2 \parallel$ is norm of D2. Given the vectors, the cosine similarity between D1 and D2 is 1, indicating they are 100% similar. Again, this 100% similarity is only based on '*protect*' and '*agreement*' words.

For another example, compute the TF-IDF values for the descriptors '*health-related*' and '*climate-related*' across each aforementioned sentence. Considering only the adjectives, each term appears once in its respective sentence and does not appear in the others. Since each sentence has six terms, the TF for each adjective in its respective sentence is 0.167, but in other sentences, it is 0. The IDF is log(2) for each adjective term since each term appears in just one sentence. The vectors for D1 and D2 based on our selected terms are as follows,

$$Vector\ for\ D1:\ [0.167 \times log(2), 0]$$

$$Vector\ for\ D2:\ [0, 0.167 \times log(2)]$$

Given that D1 and D2 are orthogonal, the dot product for these vectors is 0. Thus, the cosine similarity between D1 and D2 is 0, indicating they are entirely dissimilar based on these two selected terms (i.e., *health-related* and *climate-related*).

It is crucial to pre-process the BIT clauses before calculating text similarity using the TF-IDF and cosine similarity approaches. There are a few steps for the pre-processing. Tokenisation, a fundamental method in natural language processing, involves breaking down the text into its constituent elements, commonly referred to as tokens or words [17], which not only aids in comprehending the grammatical structure of clauses but also in determining the semantics and underlying ideas. The second step is eliminating stop terms (e.g., comma, semicolon and invited comma) since they are typically judged uninformative [18]. While researchers commonly use these terms in literature, they usually add nothing to the content or context. By removing these terms, we can concentrate on the crucial keywords and phrases that give essential insight into the substance. The last step is stemming and lemmatisation. Stemming is the process of reducing words to their most basic form or root, which might result in non-standard or non-dictionary terms [19]. For example, the words '*investigator*' and '*investigation*' should be reduced to '*invest*', according to stemming. Lemmatisation is a more grammatically accurate method of reaching a similar goal by reducing words to their dictionary base form [20]. Stemming would return '*car*' for the word '*caring*'. Lemmatisation can ensure the return of '*care*' instead of '*car*' in this context.

## Step 2: Similarity values based on a given threshold

The second step involves extracting the first $M$ values based on a given threshold value ($t$) out of the $^{N}C_2$ text-similarity scores from the first step. All scores $\geq t$ will be counted for this extraction. These $M$ values are associated with $2M$ documents or BITs. The selected text-similarity value for $t$ is usually very high to ensure the inclusion of documents or clauses revealing high similarity with others.

## Step 3: Distinct documents and their frequency and weighted frequency

Within the $2M$ documents or BITs (from the second step), some appear only a few times, while others seem more frequently. Let's assume that there are $D$ distinct documents or BITs

within these 2*M* documents. We then quantify the frequency of each of these *D* documents. If these frequency values are $w_1, w_2, w_3 \ldots \ldots \ldots w_D$, then their sum equals 2*M*. For normalisation, we then divide each frequency value by 2*M*, which ensures that the final *INF* value for each country will remain within a range between 0 and 1.

## Step 4: *INF* calculation for each country

From the third step, we have *D* distinct documents or BITs and their weighted frequency values. We need to look back to the original dataset to extract the two country names (country1 and country2) associated with each of these *D* documents. In total, there will be 2*D* country names. Like documents or BITs, some appear more frequently than others and vice versa. Once again, assume there are *C* unique countries in these 2*D* countries. To capture the *INF* value for a given country $C_i$ from *C*, we sum all weighted frequency values of each of the *D* documents or BITs only if that country name appears either in country1 or country2. Mathematically,

$$INF_{C_i} = \sum_{i=1}^{D} w_i; \; \forall_{C_i} \; in \; D_i \tag{2}$$

The country that shows the highest $INF_{C_i}$ value is the country that tends to play the most critical role in disseminating clauses within the given corpus and vice versa. Since the given corpus is based on a particular BIT type and period, it can be concluded that that country dominated the clause dissemination for the underlying BIT type within the period.

Fig 1 provides a comprehensive illustration of these four steps with an example based on some of the research data used in this study. For the simplicity of this illustration, we only consider the outcome from the first step.

**Minimum and maximum *INF* values.** A country's minimum and maximum *INF* values in a given dataset can be 0 and 1, respectively. If the similarity scores of a country's BIT text with others are lower than the given threshold value, that country will have an *INF* score of 0. In Fig 1, Spain has an *INF* score of 0 since none of the BIT pairs extracted in the second step has Spain as a country in its two country columns. The minimum score mainly depends on the threshold value. A threshold value of 1 (or 100% similarity) for the data in Fig 1 will lead to an *INF* value of 0 for Nigeria, Singapore and Spain.

If each BIT of the selected BIT pairs in the second step of Fig 1 has a common country in their country columns, then that country will have an *INF* score of 1. For a threshold value of 1 in Fig 1, the selected unique BITs in the third step of Fig 1 will be 1, 3 and 4; each has Canada in common as a country. In that case, Canada will have an *INF* score of 1. Like the minimum value, the maximum *INF* score depends on the threshold value. Regardless of the threshold value, the *INF* score for a country can be 1. If each BIT of the dataset has a common country, then that country will have a threshold value of 1 for any given threshold value. The corresponding network diagram of those BITs will be a star network, as in Fig 2. This figure is based on six 'offensive' treaties from the research dataset between March 2018 and January 2020. In this network diagram, each edge is labelled with the date a BIT was signed between two countries represented by its two end nodes. Brazil will have an *INF* score of 1 for any threshold value of the similarity score.

The considered threshold value has a direct impact on the resultant *INF* score. A lower threshold value will lead to more BITs in the third step. Thus, the weighted frequency values will be smaller since the total number of BITs is used as the denominator to normalise this value, eventually lowering the resultant *INF* values for any country. Overall, the threshold value is proportional to the final *INF* scores for each country. When the threshold value is higher, the *INF* score also tend to be higher and vice versa.

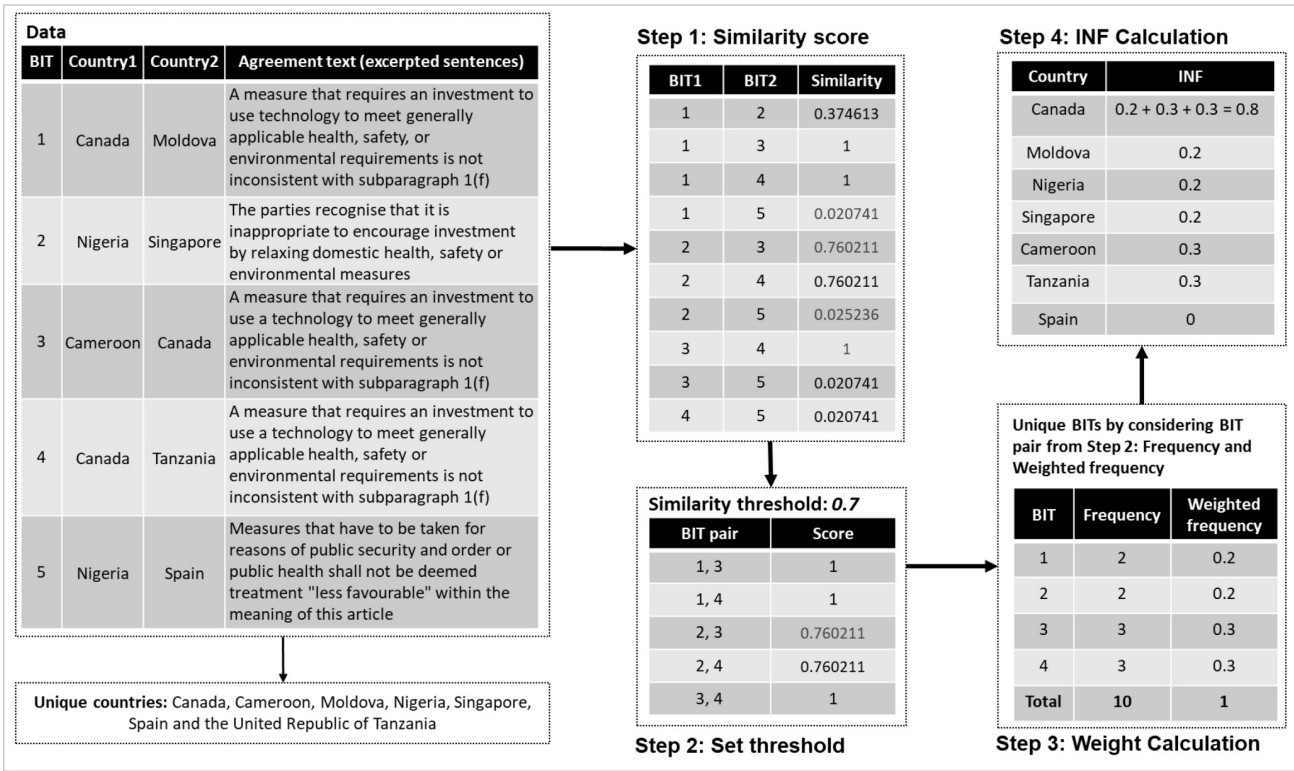

**Fig 1. Illustration of the *INF* calculation based on five Bilateral Investment Treaties (BITs).** This selection of BIT data can be guided by different criteria, such as timeframe (e.g., between two years) and among a set of BITs from a particular region. Canada has been found to play the most significant role in disseminating BIT texts. Step 2 considered a similarity threshold of 0.7. A different threshold consideration would lead to varying findings in Step 4. Spain has an INF score of 0 since none of the BITs in Step 3 has Spain in its country columns of the data (left-most table).

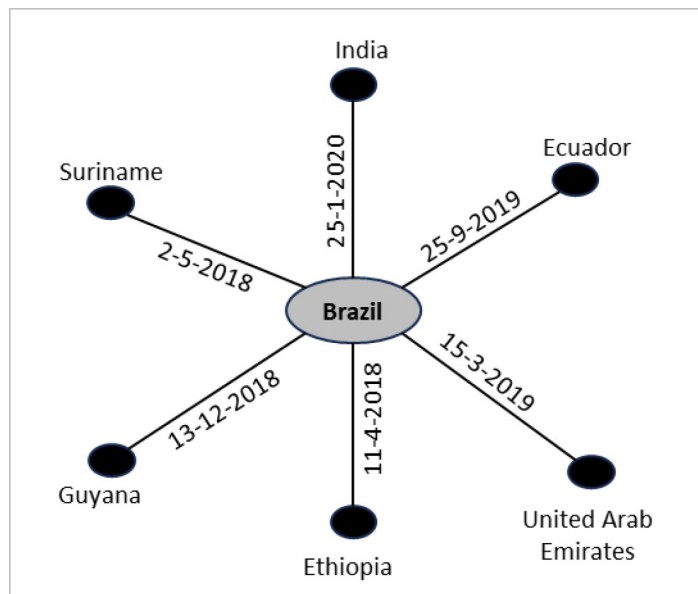

**Fig 2. Illustration of a network diagram based on six BITs containing offensive health safeguards where a country (Brazil) has an *INF* score of 1.**

**Identifying influential country/ies in different contexts using *INF*.** The proposed *INF* measure can reveal the influential country/ies in various contextual settings. The data selection is vital to draw this contextual setting. For example, to identify country/ies playing a significant role in '*defensive*' clause dissemination between 2000 and 2010, we need to select '*defensive*' BITs signed between 2000 and 2010. Similarly, we need to extract all '*offensive*' BITs to explore a country's role in spreading relevant clauses among such BITs.

## 3. Application of the proposed *INF* measure

This section illustrates the *INF* application in identifying which country played a significant role in dispersing IIA texts for defensive, neutral and offensive IIAs over time. We utilise the Electronic Database of Investment Treaties (EDIT) for Bilateral Investment Treaty (BIT) data [21], extracting and categorising information from 3298 treaties that contained the term '*health*' in contexts excluding unrelated ones. Health provisions were initially prevalent, first appearing in the 1959 Germany-Pakistan BIT, but decreased in the 1970s and 1990s until resurging in the 2000s, with over 60% of signed BITs referencing health (although noting there were fewer agreements signed in this period). According to the previous EDIT analysis, 18% (584 out of 3298) of BITs contain 934 distinct health mentions [3]. We excluded 27 uncategorised BITs and two BITs involving more than two countries or regions from 584 BITs for analysis.

Consequently, we used 555 BITs for this research. Of these 555 BITS, 151 were classified into more than one of the three categories (12 BITs into three and 125 into two categories). The multiclass BITs are considered in each class, resulting in 419 BITs containing defensive health safeguards, 263 having neutral health safeguards, and 22 containing offensive health safeguards.

### Descriptive statistics of the similarity values

There are 87,571, 34,453 and 231 similarity pairs ($^{N}C_2$) for the 419 defensive, 263 neutral and 22 offensive BITs, respectively. Table 1 details the basic statistics of these similarity values, measured according to the previously described first step of the *INF* calculation. The mean values vary slightly between categories, with neutral BITs (0.3609) scoring the highest, followed by defensive (0.3115) and offensive (0.2516). The standard deviations also have a narrow range, indicating a high level of consistency within each mentioned category. In addition, the data covered the entire theoretical range from 0 to 1 for both defensive and neutral strategies, with the maximum value for offensive being slightly less than 1 (0.9941). Each category's 25%,

**Table 1. Summary statistics of BITs and similarity values.**

| Item | Defensive | Neutral | Offensive |
|---|---|---|---|
| Number of BITs | 419 | 263 | 22 |
| Number of BIT pair | 87571 | 34453 | 231 |
| Mean similarity value | 0.3115 | 0.3609 | 0.2516 |
| Standard deviation of similarity value | 0.2757 | 0.2369 | 0.2615 |
| Minimum similarity value | 0 | 0 | 0 |
| 25% quantile of similarity value | 0.0839 | 0.1750 | 0.0839 |
| Median similarity value | 0.2375 | 0.3097 | 0.1417 |
| 75% quantile of similarity value | 0.4460 | 0.5291 | 0.2966 |
| Maximum similarity value | 1 | 1 | 0.9941 |

50%, and 75% percentile values further emphasise the unique distribution differences, providing valuable insights into the varied health mentions.

### *INF* for top countries

Table 2 outlines the *INF* values for top countries at different periods based on a threshold value of 0.80. Governments first signed BITs containing offensive health safeguards during 2010–2019. No government signed a neutral BIT during the first three decades since 1960. Of the six periods, Germany is at the top position four times for the defensive category except for 1990–1999 and 2010–2019. Brazil has the highest score for 2010–2019, when 19 BITs containing offensive health safeguards were signed. Three more BITs with offensive health safeguards were signed in 1988, 1990 and 1994. Due to this small number, the proposed measure cannot calculate the *INF* score for 1980–1989. Although we can figure this score for 1990–1999, it will not make good sense for interpretation. For these reasons, we did not report *INF* scores in Table 2 for 1980–1989 and 1990–1999. Turkey has become the top country in the recent decade for BITS with defensive and neutral health safeguards, with *INF* scores of 0.2740 and 0.3196, respectively, indicating its dominating role in IIA text dissemination in recent times.

Table 3 summarises the *INF* values for the top 10 countries considering the entire dataset from 1959 to 2021. A value of 1 has been chosen as the threshold value for the defensive and neutral categories and 0.9 for the offensive one, as there is no instance of 1 for this category. Germany leads with an *INF* of 0.6381 in the defensive category, followed by Mauritius (0.0908) and Singapore (0.0800). Canada stands out in the neutral category, with the highest *INF* of 0.4239, trailed by the Belgium-Luxembourg Economic Union (0.2076) and the United States of America (0.1246). Lastly, Brazil holds the top position in the offensive category with an *INF* of 0.9091, significantly ahead of the rest. Other countries like Turkey and Ghana rank lower in the offensive category. This distribution reflects dynamic geopolitical interaction, with nations exhibiting distinct influence and strategic orientation patterns.

This study then explores the *INF* value changes for the top 10 countries based on the entire dataset considered in this study. Table 4 details the longitudinal changes in *INF* values for the top 10 countries of Table 3 in different categories. Only one country (Germany) in the defensive category reveals an *INF* score for each period. Results also show that countries ranked in the top 10 for *INF* during the past six decades (Table 3) have taken different rank positions in each decade analysed (Table 4). No country in the neutral and offensive category shows an *INF* score for each period. Most cells of these two categories (Table 4(B) and 4(C)) do not have a score. This shows countries mostly started using neutral and offensive mentions after 2000. Only Jordan has an *INF* score during 1990–1999 for the neutral category. It is Bangladesh that has an *INF* score for 2000–2009. No other country had an *INF* score before 2000 for the offensive category.

**Table 2. *INF* values (inside the bracket) for top countries at different periods (threshold value = 0.80).**

| Period | Defensive | Neutral | Offensive |
|---|---|---|---|
| 1960–1969 | Germany (0.1173) | - | - |
| 1970–1979 | Germany (0.0698) | - | - |
| 1980–1989 | Germany (0.0651) | - | - |
| 1990–1999 | Germany (0.0997) | USA (0.0542) | - |
| 2000–2009 | Germany (0.1264) | Finland (0.2673) | - |
| 2010–2019 | Turkey (0.1462) | Turkey (0.2805) | Brazil (0.8824) |

An "-' means insufficient data to calculate the *INF* score. USA for the United States of America.

**Table 3. Top 10 countries and their *INF* values for different categories based on the entire dataset.** Only nine countries engaged in offensive BITs. USA: United States of America; UAE: United Arab Emirates and BLEU: Belgium-Luxembourg Economic Union.

| Defensive | | Neutral | | Offensive | |
|---|---|---|---|---|---|
| *Country* | *INF* | *Country* | *INF* | *Country* | *INF* |
| Germany | 0.6381 | Canada | 0.4239 | Brazil | 0.9091 |
| Mauritius | 0.0908 | BLEU | 0.2076 | Ecuador | 0.1818 |
| Singapore | 0.0800 | USA | 0.1246 | UAE | 0.1818 |
| Canada | 0.0658 | Finland | 0.0900 | Guyana | 0.1818 |
| Kuwait | 0.0433 | Turkey | 0.0433 | Suriname | 0.1818 |
| China | 0.0419 | Kosovo | 0.0415 | Ethiopia | 0.1818 |
| Egypt | 0.0411 | Cameroon | 0.0346 | Turkey | 0.0909 |
| Ethiopia | 0.0278 | Jordan | 0.0329 | Bangladesh | 0.0455 |
| Oman | 0.0278 | Guinea | 0.0311 | Ghana | 0.0455 |
| Yemen | 0.0278 | Mauritius | 0.0294 | - | - |

## Selection of the threshold value

As mentioned in the second step, selecting the threshold value is essential in measuring the *INF* value for a given dataset. Here, we suggest a retrospective approach for determining this threshold value based on the distribution of similarity values. A common understanding is that there is a higher chance of dissemination influence if two BIT texts have a higher similarity score and vice versa. The underlying phenomenon is picking up a value as the threshold more elevated than most similarity values, such as 80%, 85%, 90% and 95% of all scores. For this, we first need to observe the distribution of the similarity scores for different categories. Below, we describe how a threshold value can be measured against different target values for different IIA categories.

We first draw the cumulative distribution chart for all similarity scores for each category (Fig 3). The x-axis represents the similarity score, while the y-axis denotes the frequency of each BIT pair. All the histograms are skewed to the right, indicating a tail on the right side of the distribution. This means that the majority of the similarity score points fall on the left side of the histogram. Therefore, analysing the tail is beneficial as it allows us to identify BIT pairs with high similarity and determine which country plays a significant role in disseminating IIA text. For the tail section of this chart, we then pick up a score higher than most (e.g., 95% or 100%) of the values. For our entire research dataset, threshold values of 0.90, 0.70 and 0.90 are higher than 95% of all similarity values for the defensive, neutral and offensive categories, respectively. For example, out of the 87,581 defensive pairs, only 5% of pairs (i.e., 4,378) have a 95% or above similarity score.

The same approach can be used for any other dataset or subset of our dataset.

## Software tool for identifying *INF* value

We developed a software tool based on our proposed four-step *INF* calculation approach using Python and PowerBI. This tool can be accessed from https://inftool.com/. Anyone can access this tool from this weblink. This tool has been given access to our research data for making illustrations. A demonstrative video on how to use this tool can be found at http://video.inftool.com.

**Table 4. Longitudinal changes in *INF* values for the top 10 countries of Table 3 in different categories.** A hyphen (-) indicates that the corresponding country was not involved in any BITs during the underlying period. Only nine countries engaged in offensive BITs. BLEU stands for the Belgium-Luxembourg Economic Union.

(a) Defensive category

| Country | 2010–2019 | 2000–2009 | 1990–1999 | 1980–1989 | 1970–1979 | 1960–1969 |
|---|---|---|---|---|---|---|
| Germany | 0.0024 | 0.0549 | 0.0587 | 0.0334 | 0.031 | 0.0644 |
| Mauritius | 0.0119 | 0.0382 | 0.0167 | - | 0.0024 | - |
| Singapore | 0.0143 | 0.0119 | 0.0191 | 0.0024 | 0.0024 | - |
| Canada | 0.0477 | 0.0119 | 0.0406 | - | - | - |
| Kuwait | 0.0072 | 0.0167 | 0.0215 | 0.0024 | - | - |
| China | 0.0191 | 0.0119 | 0.0024 | 0.0119 | - | - |
| Egypt | 0.0024 | 0.0072 | 0.0048 | - | 0.0072 | - |
| Ethiopia | 0.0072 | 0.0024 | - | - | - | 0.0024 |
| Oman | - | 0.0024 | - | - | 0.0024 | - |
| Yemen | - | 0.0024 | - | - | 0.0024 | - |

(b) Neutral category

| Country | 2010–2019 | 2000–2009 | 1990–1999 | 1980–1989 | 1970–1979 | 1960–1969 |
|---|---|---|---|---|---|---|
| Canada | 0.0608 | 0.019 | - | - | - | - |
| BLEU | 0.0076 | 0.076 | - | - | - | - |
| USA | - | 0.0076 | - | - | - | - |
| Finland | - | 0.1065 | - | - | - | - |
| Turkey | 0.1369 | - | - | - | - | - |
| Kosovo | 0.019 | - | - | - | - | - |
| Cameroon | 0.0114 | - | - | - | - | - |
| Jordan | 0.0114 | 0.0152 | 0.0038 | - | - | - |
| Guinea | 0.0076 | - | - | - | - | - |
| Mauritius | 0.0114 | 0.0114 | - | - | - | - |

(c) Offensive category

| Country | 2010–2019 | 2000–2009 | 1990–1999 | 1980–1989 | 1970–1979 | 1960–1969 |
|---|---|---|---|---|---|---|
| Brazil | 0.4545 | - | - | - | - | - |
| Ecuador | 0.0455 | - | - | - | - | - |
| UAE | 0.1364 | - | - | - | - | - |
| Guyana | 0.0455 | - | - | - | - | - |
| Suriname | 0.0455 | - | - | - | - | - |
| Ethiopia | 0.0455 | - | - | - | - | - |
| Turkey | 0.0909 | - | - | - | - | - |
| Bangladesh | 0.0909 | 0.0455 | - | - | - | - |
| Ghana | 0.0455 | - | - | - | - | - |

## Evaluation of the *INF* measure

Tables 2 and 3 show that some countries stand out as pioneering countries, including Germany, Canada, the USA, Turkey, and Brazil, in BITs. For the evaluation purpose of the proposed *INF* measure, we first qualitatively review the current literature to explore countries playing a significant role in IIA text dissemination to check whether or not the *INF* reveals the same.

Germany was one of the first countries to adopt a BIT, which included a health safeguard. In the literature, Germany has been identified as a pathfinder, impacting the structure and substance of many later BITs globally [6]. Many countries have since accepted benchmarks set by the nation's historically thorough and careful approach to BITs [22]. In parallel, Canada's Foreign Investment Promotion and Protection Agreements have been echoed in global investment protocols, with the North American Free Trade Agreement's Chapter 11 serving as a compass to the development of this agreement [23, 24]. Turkey created BITs that integrated

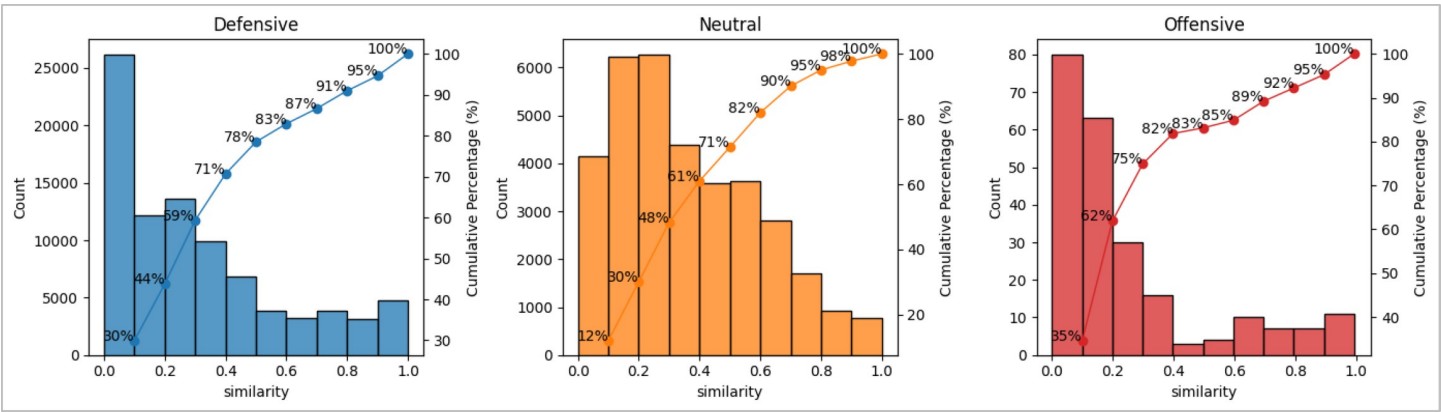

**Fig 3. Cumulative distribution chart for text similarity for defensive, neutral and offensive categories.**

Eastern and Western ideas due to its particular geographic and cultural circumstances, carving out a role for itself as a leader in investment treaty discussions [25]. Brazil developed The Brazilian Cooperation and Facilitation Investment Agreement, an innovative IIA model that contains a series of offensive health safeguards and does not feature the divisive investor-state dispute settlement processes; this approach has made Brazil a pioneer in shaping global IIA practice [26, 27]. Overall, the proposed *INF* measure is consistent with the existing literature regarding the countries identified as influential concerning health safeguards in BITs. This summary review suggests that the *INF* is effective for quantification of the complex dynamics of BIT influence.

## 4. Discussion

The study proposes a new metric, influence (*INF*), to assess the involvement of different countries or regions in disseminating IIA clauses over time. We also demonstrated its application in identifying influential countries that significantly spread clauses among BITs that explicitly mention health provisions. This metric can also be applied to identify leading countries that disseminate IIA clauses for other issues, such as climate change. This innovation is vital to expand the understanding of global economic governance and trade relations dynamics by showcasing the diffusion of global investment protection concepts, norms, and standards and their drivers. This metric also helps predict trends in international investment law and policy.

This study suggested a retrospective approach for determining the threshold value for different ceilings (e.g., a ceiling score higher than 80% of all scores) in the second step of the *INF* calculation. A selected good-range threshold value ensures to exclude countries having BITs with many countries but with lower text-similarity scores among clauses. For example, a government could engage in many BITs, but their text similarity scores with other BITs are low. The threshold score in the second step will ensure a low *INF* score for that country, although it has many BITs with others. However, there is still an open-ended question—which ceiling score should we consider for a given set of BITs? A higher ceiling score will leave less number of BITs in the *INF*'s third step and vice versa. If we have fewer BITs, more countries of the dataset will have a 0 value in the *INF*'s fourth step, which will not be a desirable outcome, especially if we want to compare the prominence of IIA text dissemination among countries. A lower ceiling score would undermine the differences in similarity scores between BIT pairs. For example, assume we have scores for 90 similarity pairs (0.01, 0.02, 0.03. . . .0.90). Now, if we select a ceiling score that is higher than 60% of all similarity scores (i.e., 0.54), it will put BIT

pairs with a score of 0.54 and 0.90 in the same group although there is a significant difference (0.90–0.54 = 0.36) between them. Future research with different dataset sizes and BITs from other provisions, such as tobacco use and climate change, could address this issue and provide a suggestive guideline for selecting this ceiling value for different IIA contexts.

This study considered TF-IDF for quantifying the similarity between BIT texts in the first step of the *INF* calculation. However, TF-IDF does not account for the semantic meaning of words, which is crucial in legal documents where identical terms may have different implications depending on their usage [28]. Moreover, this method is sensitive to document length, which could lead to a skewed similarity assessment in lengthier clauses. There are other potential candidates to overcome these limitations, such as Bidirectional Encoder Representations from Transformer (BERT) [29] and Word2Vec [30]. We also did other experiments for similarity calculations using these two approaches (data not shown) and did not find many differences in the corresponding similarity values. Accordingly, the resulting *INF* values do not differ much across these three similarity approaches. The size of the BIT agreement corpus would not be unexpectedly large due to the smaller number of countries worldwide, a potential reason for not showing up different *INF* values for different similarity approaches. Likewise, although we used the cosine similarity with potential limitations in recognising semantic similarities [14], many other candidates for the similarity measure (e.g., Euclidean distance and Manhattan distance) exist. The size of the BIT agreement corpus would not be unexpectedly large due to the smaller number of countries worldwide. For this reason, this study used simple and easily interpretable TF-IDF and cosine similarity methods in the first step of the proposed *INF* measure. Future research could compare these alternative candidates to select the best ones.

The proposed *INF* metric scored high for major countries or regions consistently shaping health-related BIT provisions. According to this measure, Germany, Canada, and Brazil emerged as the most influential players in defensive, neutral, and offensive health mentions, respectively. These countries wield substantial bargaining power in international investment law and policy, and their innovative approaches to BITs set a path for others to follow. Recognising these countries provides crucial insights into international investment regulations' direction and sources of influence. Policymakers might benefit from observing these leading nations' practices and evaluating the potential impact of adopting similar strategies in their BIT negotiations.

Furthermore, analysing text similarity and *INF* scores over several periods reveals intriguing trends. This analysis divides the period into 2000 to 2010 and 2011 to 2020. Germany's sustained influence in defensive health clauses during both periods indicates a consistent approach to health-related provisions. Canada's ascent to the top spot in neutral health references from 2011 to 2020 demonstrates a shift in the country's policy focus towards health initiatives during this decade. The rise in offensive health safeguards between the two periods reflects changes in international dynamics and Brazil's proactive approach to employing BITs to advance a public health agenda. These findings underscore the importance of monitoring and analysing changes in health clauses over time to understand governments' evolving priorities and potential shifts in IIAs. These findings also highlight an opportunity for qualitative research to explore further the policy processes and political economy dynamics that influence these trends [2]. There may also be important geopolitical and regional dynamics, which could helpfully be explored in future research.

The findings from applying the *INF* metric to BITs mentioning health provision carry significant implications for policymakers and investors. Understanding the evolving roles of health provisions in BITs enables governments to devise more effective IIAs that strike a balance between fostering investment and safeguarding public health [3, 31]. Moreover, policymakers can learn from influential countries or regions, like Germany, Canada, and Brazil, to

chart new directions in their BIT negotiations and also further innovate and improve the safeguards for health. Conversely, recognising the shifting landscape of health provisions in BITs can benefit investors. Understanding some countries' aggressive stances towards health-related investment projects may assist investors in identifying possible opportunities and aligning their investment strategies accordingly.

## 5. Conclusion

Our proposed *INF* measure can effectively identify countries that have played a significant role in clause dissemination within BITs. Our findings can aid policymakers and investors in navigating the complexities of IIAs and their impacts on public health and economic growth. Countries can gain predictive power and strengthen their negotiating positions by utilising their *INF* scores, leading to more effective and balanced IIAs. Furthermore, our findings offer a deeper look into the dynamics of international economic governance and trade relationships, particularly in promoting investment and safeguarding public health.

While providing a comprehensive snapshot of current BIT patterns, our text similarity analysis also points to areas for further research. This includes exploring how health safeguards are globally disseminated into IIAs, emphasising newly introduced areas such as climate change and environmental agreements. We also recognise the importance of integrating a finer discussion of the geopolitical and economic contexts, especially those influencing BIT practices in the Asia-Pacific region. This region, often at the forefront of public health challenges like tobacco control, provides a critical backdrop for understanding the intersection of public policy and health-related exceptions in BITs. Acknowledging the limitations of our methodology, we suggest that future research should delve into these contextual factors to provide a more comprehensive understanding of BIT practices and their implications.

To conclude, our study contributes to the existing literature by providing a novel method to analyse BITs and paves the way for future investigations. These should blend NLP findings with an in-depth analysis of geopolitical and economic influences, particularly in regions where public policy concerns are deeply intertwined with health-related exceptions in BITs. Such scholarly exploration would enhance our understanding of international investment law and foster a more holistic approach to balancing economic development and public health protection.

## Author Contributions

**Conceptualization:** Shahadat Uddin.

**Formal analysis:** Shahadat Uddin, Haohui Lu.

**Funding acquisition:** Shahadat Uddin, Anne Marie Thow.

**Methodology:** Shahadat Uddin, Haohui Lu, Wolfgang Alschner.

**Software:** Shahadat Uddin, Haohui Lu.

**Supervision:** Shahadat Uddin.

**Writing – original draft:** Shahadat Uddin, Haohui Lu, Wolfgang Alschner, Dori Patay, Nicholas Frank, Fabio S. Gomes, Anne Marie Thow.

**Writing – review & editing:** Shahadat Uddin, Haohui Lu, Wolfgang Alschner, Dori Patay, Nicholas Frank, Fabio S. Gomes, Anne Marie Thow.

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
