## [Decision Letter · Decision Letter 0]

10 Dec 2023

PONE-D-23-40343

An NLP-based Novel Approach for Assessing National Influence in Clause Dissemination across Bilateral Investment Treaties

PLOS ONE

Dear Dr. Uddin,

Thank you for submitting your manuscript to PLOS ONE. After careful consideration, we feel that it has merit but does not fully meet PLOS ONE’s publication criteria as it currently stands. Therefore, we invite you to submit a revised version of the manuscript that addresses the points raised during the review process.

We look forward to receiving your revised manuscript.

Kind regards,

Poowin Bunyavejchewin

Academic Editor

PLOS ONE

Journal Requirements:

   "National Health and Medical Research Council (NHMRC, Government of Australia), Grant ID 2012233." 

Additional Editor Comments:

Although everything appears academically sound, the reviewers have suggested that some revisions be made. Please review their comments and consider revising accordingly. Reviewer 1 has requested another round of review following the submission of a revised manuscript.

Reviewers' comments:

Reviewer's Responses to Questions

**Comments to the Author**

1. Is the manuscript technically sound, and do the data support the conclusions?

Reviewer #1: Yes

Reviewer #2: Yes

2. Has the statistical analysis been performed appropriately and rigorously? 

Reviewer #1: Yes

Reviewer #2: Yes

3. Have the authors made all data underlying the findings in their manuscript fully available?

Reviewer #1: Yes

Reviewer #2: Yes

4. Is the manuscript presented in an intelligible fashion and written in standard English?

Reviewer #1: Yes

Reviewer #2: Yes

5. Review Comments to the Author

Reviewer #1: The manuscript investigates the influence of national policies in bilateral investment treaties, particularly regarding health provisions. It leverages Natural Language Processing (NLP) to analyze the text similarity of treaties to determine how countries propagate their investment policies. This topic is timely and significant given the growing concern about the intersection of international investment law and public health policies.

The authors propose an ‘Influence’ metric (INF) developed through NLP-based text similarity analysis of BITs/IIAs. They analyze the propagation of specific clauses across treaties and identify the key countries that shape international investment agreements. The study finds that countries such as Germany, Canada, and Brazil are influential in spreading defensive, neutral, and offensive health-related clauses in BITs, respectively. These insights are crucial for policymakers and investors to understand the dynamics of global IIA text dissemination.

First, let me mention that there are a few typos and infelicities that should be corrected. I have identified typos and grammatical errors in the draft as follows:

• “For this reason including” should be “For this reason, including” (line 65).

• “However the existing literature” should be “However, the existing literature” (line 68).

• “which can provide” should be “which provides” (line 72).

• “According to the INF Germany Canada and Brazil” should be “According to the INF, Germany, Canada, and Brazil” (line 76).

• “and their innovative approaches to BITs set a” should be “and their innovative approaches to BITs, set a” (line 79).

• “It can help them identify” should be “This can help them identify” (line 82).

• “In response there has” should be “In response, there has” (line 613).

Here are my comments on substance:

1. In the study’s first step, which employs TF-IDF and cosine similarity to analyze the textual similarity in bilateral investment treaties, there are inherent methodological constraints. In particular, TF-IDF’s inability to discern the contextual nuances in legal documents, where identical terms may have different implications depending on their usage, and cosine similarity’s limitation in recognizing semantic similarities, crucial in legal texts where synonyms and polysemy are prevalent, is noteworthy. Moreover, the sensitivity of the TF-IDF method to the length of documents could lead to skewed similarity assessments, especially in lengthier clauses, suggesting that a more sophisticated approach, possibly embedding-based models such as Word2Vec or BERT, which can comprehend both context and semantic meaning, might yield a more accurate and nuanced analysis of such legal texts. I am not saying that this should be changed, but I strongly suggest that the authors disclose and discuss potential limitations.

2. I have another concern regarding Step 4. The approach that sums the weighted frequency values of BITs to determine a country’s influence in clause dissemination could potentially oversimplify the complex geopolitical and economic relationships inherent in these treaties. This method assumes a direct correlation between the frequency of treaty involvement and influence, potentially overlooking the qualitative aspects of influence such as a country’s economic power or diplomatic reach, which might not be accurately reflected in mere frequency counts. Moreover, this quantitative approach might not account for the strategic importance or impact of specific BITs, where a country’s involvement in a few key treaties could be more influential than frequent participation in less impactful treaties. This could lead to a skewed representation of a country’s actual influence on clause dissemination, suggesting the need for a more multifaceted approach that combines quantitative frequency analysis with qualitative assessments of treaty impact and geopolitical significance.

3. This paper should engage more deeply with the literature on BITs and tobacco. Notable works include Tania Voon and Andrew Mitchell’s analysis in “Implications of international investment law for plain tobacco packaging: lessons from the Hong Kong–Australia BIT” (Public Health and Plain Packaging of Cigarettes: Legal Issues, Edward Elgar, 2012, ISBN 9780857939425), in which there is a seminal examination of the balance between international investment agreements and domestic health protections can deepen understanding of health-related clauses in BITs, a crucial aspect of the study. Julien Chaisse’s “Exploring the Confines of International Investment and Domestic Health Protections” (American Journal of Law & Medicine, 39(2-3), 332-360, https://doi.org/10.1177/009885881303900208), because This article provides a theoretical framework, comparative insights, and case studies that are directly relevant to assessing the influence of countries in the propagation of health clauses in BITs. Additionally, the article’s focus on general exception clauses can inform the study’s approach to evaluating clause dissemination, offering a more nuanced view of how such clauses are integrated, and their implications. Also, Voon and Mitchell’s “Implications of International Investment Law for Tobacco Flavouring Regulation” (The Journal of World Investment & Trade, 12(1), 65-80, https://doi.org/10.1163/221190011X00120). Engaging with this article adds a legal and policy dimension to the study, enhancing its depth and relevance.

4. Similarly, Vytiaganets’s article on tobacco regulation (The Journal of World Investment & Trade, 21(5), 753-780, https://doi.org/10.1163/22119000-12340194) and Melillo’s study on evidentiary issues (The Journal of World Investment & Trade, 21(5), 724-752, https://doi.org/10.1163/22119000-12340193) are crucial for a comprehensive understanding of the nexus between international investment law and tobacco control. Also, while reviewing this submission, I was reminded of an article I read last summer by Julien Chaisse, Manfred Elsig, Sufian Jusoh, and Andrew Lugg (2022) in the Journal of International Economic Law [Julien Chaisse, Manfred Elsig, Sufian Jusoh, Andrew Lugg, Drafting Investment Law: Patterns of Influence in the Regional Comprehensive Economic Partnership (RCEP), Journal of International Economic Law, Volume 25, Issue 1, March 2022, Pages 110–128, https://doi.org/10.1093/jiel/jgac006 ], which discussed the influence of Asian BIT practices on the RCEP, particularly regarding exception clauses relevant to health. This is significant, considering the prevalence of tobacco-related disputes in Asia-Pacific countries, which are notably concerned with health-related exceptions. This article analyzes the impact of two Japanese agreements, the CPTPP and Japan-Uruguay BIT, on the RCEP investment chapter, categorizing each article’s overlap with other agreements as high, medium, or low. The analysis revealed common trends in copy-pasting language with adjustments for specific needs, especially in articles toward the end of agreements, which often contain unique or emerging provisions.

5. Last but not the least, I think conclusion of the submission could be improved as it could incorporate a finer discussion of the geopolitical and economic contexts influencing BIT practices, especially in the Asia-Pacific region. This would enrich the interpretation of the NLP-based findings. Also, considering the limitations of the methodology and suggesting avenues for further research, particularly in areas where health-related exceptions in BITs overlap with public policy concerns, would provide a more comprehensive and insightful ending to the study (and invite future scholarly exploration).

Therefore, major revisions have been recommended. The paper presents an innovative approach with significant implications for understanding international investment law. The revisions, while substantive, should be straightforward provided that my specific comments are addressed thoroughly. I ma happy t review the revised piece in due course.

Reviewer #2: While this study presents a valuable contribution to understanding the influence of countries in shaping health provisions within International Investment Agreements (IIAs), some recommendations for further research and consideration are pertinent. Firstly, it would be beneficial to augment the quantitative approach of the INF metric with qualitative analyses. Incorporating in-depth case studies and interviews with key negotiators could provide richer insights into the diplomatic processes that influence the inclusion of health safeguards in IIAs.

Moreover, regional dynamics play a crucial role in shaping international agreements. Future research could delve deeper into how regional considerations impact a country's approach to health provisions in IIAs. This may offer a more nuanced understanding of the varying priorities and challenges faced by nations in different parts of the world.

Additionally, recognizing the evolving nature of international agreements, a longitudinal analysis could provide insights into how countries' roles in shaping health-related clauses within IIAs change over time. This temporal perspective would contribute to a more comprehensive understanding of the dynamics at play.

Also, the study could benefit from a more detailed discussion of potential limitations and challenges associated with the NLP-based approach, acknowledging the complexity of diplomatic negotiations and potential nuances missed by automated analyses.

Lastly, please re-check reference 1. Oxford University Press should be written in full, not abbreviated as OUP.

6. PLOS authors have the option to publish the peer review history of their article (what does this mean?). If published, this will include your full peer review and any attached files.

Reviewer #1: No

Reviewer #2: No

---

## [Author Response · Author response to Decision Letter 0]

23 Jan 2024

Reviewer response

An NLP-based Novel Approach for Assessing National Influence in Clause Dissemination across Bilateral Investment Treaties

We sincerely thank the reviewers and editor for their insightful suggestions and comments. Here is our response to the corrections suggested by each of them. Changes are marked in red colour in the revised main manuscript file.

Suggestions from the Editor

Comment 1: 

Please ensure that your manuscript meets PLOS ONE’s style requirements, including those for file naming. The PLOS ONE style templates can be found at 

Our response: Thank you very much for this suggestion. We paid particular attention to meeting the PLOS ONE style requirements while revising this manuscript.

Comment 2: 

Please note that funding information should not appear in any section or other areas of your manuscript. We will only publish funding information present in the Funding Statement section of the online submission form. Please remove any funding-related text from the manuscript.

Our response: We have done so in this revised manuscript. We removed the funding statement from the main manuscript. 

Comment 3: 

Thank you for stating the following financial disclosure: “National Health and Medical Research Council (NHMRC, Government of Australia), Grant ID 2012233.” 

Please state what role the funders took in the study. If the funders had no role, please state: “The funders had no role in study design, data collection and analysis, decision to publish, or preparation of the manuscript.” 

If this statement needs to be corrected, you must amend it as needed. Please include this amended Role of Funder statement in your cover letter; we will change the online submission form on your behalf.

Our response: We appreciate this suggestion. We added the details of the funder role in the cover letter of this revised submission. Here are the texts we added to the cover letter

This research has been funded by the National Health and Medical Research Council, Australia (Grant ID 2012233). The funders had no role in study design, data collection and analysis, publication decisions, or manuscript preparation.

Comment 4: 

Although everything appears academically sound, the reviewers have suggested some revisions. Please review their comments and consider revising accordingly. Reviewer 1 has requested another round of review following the submission of a revised manuscript.

Our response: We carefully reviewed their comments and suggestions and adopted them accordingly. Please see below this response letter and the highlighted texts in the main manuscript for our answers.

First reviewer

Comment 1:

First, let me mention a few typos and inflicities that should be corrected. I have identified typos and grammatical errors in the draft as follows:

 “For this reason including” should be “For this reason, including” (line 65).

 “However the existing literature” should be “However, the existing literature” (line 68).

 “which can provide” should be “which provides” (line 72).

 “According to the INF Germany Canada and Brazil” should be “According to the INF, Germany, Canada, and Brazil” (line 76).

 “and their innovative approaches to BITs set a” should be “and their innovative approaches to BITs, set a” (line 79).

 “It can help them identify” should be “This can help them identify” (line 82).

 “In response, there has” should be “In response, there has” (line 613).

Our response: Thank you for pointing out these grammatical errors. We fixed them accordingly in this revised manuscript. Please see the corresponding lines in the abstract and introduction sections of the main manuscript.

Comment 2:

There are inherent methodological constraints in the study’s first step, which employs TF-IDF and cosine similarity to analyse the textual similarity in bilateral investment treaties. In particular, TF-IDF’s inability to discern the contextual nuances in legal documents, where identical terms may have different implications depending on their usage, and cosine similarity’s limitation in recognising semantic similarities, crucial in legal texts where synonyms and polysemy are prevalent, is noteworthy. Moreover, the sensitivity of the TF-IDF method to the length of documents could lead to skewed similarity assessments, especially in lengthier clauses, suggesting that a more sophisticated approach, possibly embedding-based models such as Word2Vec or BERT, which can comprehend both context and semantic meaning, might yield a more accurate and nuanced analysis of such legal texts. I am not saying this should be changed, but I strongly suggest that the authors disclose and discuss potential limitations.

Our response: We appreciate this in-depth comment. We added further details about the limitations of TF-IDF and cosine similarity methods (lines 27-37 on page 17 and lines 1-6 on page 18) compared to Word2Vec and BERT. We also did other experiments for similarity calculations using these three approaches (TF-IDF, Word2Vec and BERT) and did not find many differences in the corresponding similarity values. Accordingly, the resulting INF values do not differ much across these three similarity approaches. The size of the BIT agreement corpus would not be unexpectedly large due to the smaller number of countries worldwide, a potential reason for not showing up different INF values for different similarity approaches. 

Comment 3:

I have another concern regarding Step 4. The approach that sums the weighted frequency values of BITs to determine a country’s influence in clause dissemination could potentially oversimplify the complex geopolitical and economic relationships inherent in these treaties. This method assumes a direct correlation between the frequency of treaty involvement and influence, potentially overlooking the qualitative aspects of influence, such as a country’s economic power or diplomatic reach, which might not be accurately reflected in mere frequency counts. Moreover, this quantitative approach might not account for the strategic importance or impact of specific BITs, where a country’s involvement in a few key treaties could be more influential than frequent participation in less impactful treaties. This could lead to a skewed representation of a country’s influence on clause dissemination, suggesting the need for a more multifaceted approach combining quantitative frequency analysis with qualitative assessments of treaty impact and geopolitical significance.

Our response: Thank you for this comment. Our approach sums the weighted frequency values of BITs to determine a country’s influence in clause dissemination. However, we considered a threshold value (t) in selecting these BITs in the second step. This will ensure excluding countries frequently engaged in BITs with other countries but with lower text-similarity scores between BIT documents. For example, a government could engage in many BITs, but their text similarity scores with other BITs are low. The threshold score in the second step will ensure a low INF score for that country, although it has many BITs with others. Our proposed measure assumes a direct correlation between the frequency of treaty involvement and influence; however, only for BITs that show a text-similarity score higher than a given threshold score (first two steps of the proposed measure). We did not make such an assumption for all BITs. Please see lines 12-15 on page 17 for further details. We further clarified the second step of the proposed INF measure (pages 7 and 8). 

 Our study’s primary aim is to quantitatively propose an influence metric to provide insights into the role of different countries or regions in the propagation of IIA texts among BITs. Such a ranking could open up new research opportunities that could be addressed qualitatively and quantitatively in future research endeavours. For example, as mentioned by the reviewer, a country could have very few BITs but with very high strategic importance. The proposed INF measure hence gives that country a low score. This will create new qualitative research scopes, such as, a case study-based research design targeting that particular country for exploring the strategic importance of its BITs and low INF score. We have ongoing research towards this direction with five qualitative case studies investigating why and how countries adopt health safeguards in IIAs. By applying an in-depth policy and governance analysis, our research team will identify the institutional (governance and political structures, ideational (paradigms and norms) and interest-based (political economy and influence) factors that shape countries’ approach to health safeguards. The qualitative methods applied include in-depth interviews with key informants and document analysis, informed by political science and legal theories. The quantitative component of this research project informed the case study selection for the qualitative component. For example, these case studies will investigate 1) policy learning post-Philip Morris investor-state dispute settlement cases in Australia and Uruguay; 2) innovative clauses adopted by Bangladesh that oblige investors to compensate for the public welfare damage they cause; and 3) IIA policy innovation in Latin America. This qualitative research, in combination with the quantitative component of this research project, enables this research project to provide high-quality and high-validity insights to researchers and policymakers about the factors that enable the adoption of health safeguards. 

 Similarly, our other ongoing machine learning-based study explores what country-level factors (e.g., gross domestic product, economic power, diplomatic reach, inflation rate, population and human development index) are associated with a high or low INF score. 

Comment 4:

This paper should engage more deeply with the literature on BITs and tobacco. Notable works include Tania Voon and Andrew Mitchell’s analysis in “Implications of international investment law for plain tobacco packaging: lessons from the Hong Kong–Australia BIT” (Public Health and Plain Packaging of Cigarettes: Legal Issues, Edward Elgar, 2012, ISBN 9780857939425), in which there is a seminal examination of the balance between international investment agreements and domestic health protections can deepen understanding of health-related clauses in BITs, a crucial aspect of the study. Julien Chaisse’s “Exploring the Confines of International Investment and Domestic Health Protections” (American Journal of Law & Medicine, 39(2-3), 332-360,

https://doi.org/10.1177/009885881303900208), because This article provides a theoretical framework, comparative insights, and case studies that are directly relevant to assessing the influence of countries in the propagation of health clauses in BITs. Additionally, the article’s focus on general exception clauses can inform the study’s approach to evaluating clause dissemination, offering a more nuanced view of how such clauses are integrated and their implications. Also, Voon and Mitchell’s “Implications of International Investment Law for Tobacco Flavouring Regulation” (The Journal of World Investment & Trade, 12(1), 65-80, https://doi.org/10.1163/221190011X00120). Engaging with this article adds a legal and policy dimension to the study, enhancing its depth and relevance.

Our response: Thank you for such an insightful suggestion. We have reviewed these articles and referred to them in the revised manuscript. Please see lines 3-24 on page 4 for further information.

Comment 5:

Similarly, Vytiaganets’s article on tobacco regulation (The Journal of World Investment & Trade, 21(5), 753-780, https://doi.org/10.1163/22119000-12340194) and Melillo’s study on evidentiary issues (The Journal of World Investment & Trade, 21(5), 724-752, https://doi.org/10.1163/22119000-12340193) are crucial for a comprehensive understanding of the nexus between international investment law and tobacco control. Also, while reviewing this submission, I was reminded of an article I read last summer by Julien Chaisse, Manfred Elsig, Sufian Jusoh, and Andrew Lugg (2022) in the Journal of International Economic Law [Julien Chaisse, Manfred Elsig, Sufian Jusoh, Andrew Lugg, Drafting Investment Law: Patterns of Influence in the Regional Comprehensive Economic Partnership (RCEP), Journal of International Economic Law, Volume 25, Issue 1, March 2022, Pages 110–128, https://doi.org/10.1093/jiel/jgac006 ], which discussed the influence of Asian BIT practices on the RCEP, particularly regarding exception clauses relevant to health. This is significant, considering the prevalence of tobacco-related disputes in Asia-Pacific countries, which are notably concerned with health-related exceptions. This article analyses the impact of two Japanese agreements, the CPTPP and Japan-Uruguay BIT, on the RCEP investment chapter, categorising each article’s overlap with other agreements as high, medium, or low. The analysis revealed common trends in copy-pasting language with adjustments for specific needs, especially in articles toward the end of agreements, which often contain unique or emerging provisions.

Our response: We appreciate the suggestion and have now reviewed and referred to these articles in the revised manuscript. Please see lines 3-24 on page 4 for further information.

Comment 6:

Last but not least, I think the conclusion of the submission could be improved as it could incorporate a finer discussion of the geopolitical and economic contexts influencing BIT practices, especially in the Asia-Pacific region. This would enrich the interpretation of the NLP-based findings. Also, considering the limitations of the methodology and suggesting avenues for further research, particularly in areas where health-related exceptions in BITs overlap with public policy concerns, would provide a more comprehensive and insightful ending to the study (and invite future scholarly exploration).

Our response: Thank you for this comment. We significantly revised the conclusion section (page 19). We discussed the limitations of this study and pointed out the avenues for future research directions. Please see pages 17-18 for further details.

Second reviewer

Comment 1:

Firstly, it would be beneficial to augment the quantitative approach of the INF metric with qualitative analyses. Incorporating in-depth case studies and interviews with critical negotiators could provide richer insights into the diplomatic processes that influence the inclusion of health safeguards in IIAs.

Our response: Our study’s primary aim is to quantitatively propose an influence metric to provide insights into the role of different countries or regions in the propagation of IIA texts among BITs. Such a ranking could open up new research opportunities that could be addressed qualitatively and quantitatively in future research endeavours. For example, as mentioned by the reviewer, a country could have very few BITs but with very high strategic importance. The proposed INF measure hence gives that country a low score. This will create new qualitative research scopes, such as, a case study-based research design targeting that particular country for exploring the strategic importance of its BITs and low INF score. We have ongoing research towards this direction with five qualitative case studies described above in response to Reviewer 1. In the Discussion of the revised paper, we have noted the opportunity for this quantitative research to be complemented by qualitative research approaches that engage with policy processes and political economy (Page 18, Lines 23-26).

Comment 2:

Moreover, regional dynamics play a crucial role in shaping international agreements. Future research could delve deeper into how regional considerations impact a country’s approach to health provisions in IIAs. This may offer a more nuanced understanding of the varying priorities and challenges faced by nations in different parts of the world.

Our response: We agree with your observation. In our future research endeavours, we will consider this factor (regional dynamics), along with others such as gross domestic product and population, in exploring why some countries reveal higher INF values. We have also noted the potential for this future research direction in the discussion section of the revised paper (page 18, lines 23-26).

Comment 3:

Additionally, recognising the evolving nature of international agreements, a longitudinal analysis could provide insights into how countries’ roles in shaping health-related clauses within IIAs change over time. This temporal perspective would contribute to a more comprehensive understanding of the dynamics at play.

Our response: We appreciate this comment. We did further analysis to extract the longitudinal nature of the INF value change over time. The new results are detailed in Table 4 on page 14. The corresponding textual description can be found in lines 11-17 on page 13.

Comment 4:

Also, the study could benefit from a more detailed discussion of potential limitations and challenges associated with the NLP-based approach, acknowledging the complexity of diplomatic negotiations and possible nuances missed by automated analyses.

Our response: Thank you for this comment. The revised manuscript has discussed further details on the potential limitations and challenges associated with the NLP-based approach. Please see lines 27-37 on page 17 and 1-6 on page 18 for further details. 

Comment 5:

Lastly, please re-check reference 1. Oxford University Press should be written in full, not abbreviated as OUP.

Our response: Thank you. We have corrected this reference in this revised submission.

---

## [Decision Letter · Decision Letter 1]

24 Jan 2024

An NLP-based Novel Approach for Assessing National Influence in Clause Dissemination across Bilateral Investment Treaties

PONE-D-23-40343R1

Dear Dr. Uddin,

We’re pleased to inform you that your manuscript has been judged scientifically suitable for publication and will be formally accepted for publication once it meets all outstanding technical requirements.

Kind regards,

Poowin Bunyavejchewin

Academic Editor

PLOS ONE

Additional Editor Comments (optional):

I recommend that the Editor-in-Chief accept the revised manuscript for publication in PLOS ONE.

Reviewers' comments:

Reviewer's Responses to Questions

**Comments to the Author**

1. If the authors have adequately addressed your comments raised in a previous round of review and you feel that this manuscript is now acceptable for publication, you may indicate that here to bypass the “Comments to the Author” section, enter your conflict of interest statement in the “Confidential to Editor” section, and submit your "Accept" recommendation.

Reviewer #1: All comments have been addressed

2. Is the manuscript technically sound, and do the data support the conclusions?

Reviewer #1: Yes

3. Has the statistical analysis been performed appropriately and rigorously? 

Reviewer #1: Yes

4. Have the authors made all data underlying the findings in their manuscript fully available?

Reviewer #1: Yes

5. Is the manuscript presented in an intelligible fashion and written in standard English?

Reviewer #1: Yes

6. Review Comments to the Author

Reviewer #1: After re-evaluating the revised manuscript; I am impressed with how the authors have incorporated the feedback from the initial review, including addressing methodological concerns and expanding the literature review to encompass key works like those by Voon and Mitchell. The enhancements have notably elevated the depth and quality of the research. This revised version significantly enriches our understanding of international investment law, particularly in the context of public health policies. I strongly recommend its publication in PLOS ONE

7. PLOS authors have the option to publish the peer review history of their article (what does this mean?). If published, this will include your full peer review and any attached files.

Reviewer #1: No

---

## [Editor Report · Acceptance letter]

1 Mar 2024

PONE-D-23-40343R1 

PLOS ONE

Dear Dr. Uddin, 

I'm pleased to inform you that your manuscript has been deemed suitable for publication in PLOS ONE. Congratulations! Your manuscript is now being handed over to our production team.

Kind regards, 

on behalf of

Mr. Poowin Bunyavejchewin 

Academic Editor

PLOS ONE